# Prevention of Postpartum Depression via a Digital ACT-Based Intervention: Evaluation of a Prototype Using Multiple Case Studies

**DOI:** 10.3390/bs15121723

**Published:** 2025-12-12

**Authors:** Anna Elena Nicoletti, Silvia Rizzi, Stefano Fait, Oscar Mayora-Ibarra

**Affiliations:** Digital Health Research, Centre for Digital Health & Wellbeing, Fondazione Bruno Kessler, Via Sommarive 18, 38123 Trento, Italy; annicoletti@fbk.eu (A.E.N.); sfait@fbk.eu (S.F.); omayora@fbk.eu (O.M.-I.)

**Keywords:** ACT, well-being, pregnancy, eHealth, mHealth, development, usability, user-centred design

## Abstract

Postpartum depression (PPD) affects up to 15% of mothers, yet access to preventive psychological interventions during pregnancy remains limited. Acceptance and Commitment Therapy (ACT) has demonstrated efficacy in promoting psychological flexibility and preventing mental distress. Nevertheless, no studies have yet evaluated its use for the prevention of PPD through a chatbot-based digital intervention. The present study describes the development and preliminary evaluation of an ACT-based chatbot intervention (REA) to support women during late pregnancy and the early postpartum period. Nineteen participants interacted with the low-fidelity REA prototype, explored its features, completed two questionnaires, and then participated in semi-structured interviews. Quantitative data were analysed using the Wilcoxon signed-rank test; qualitative data were analysed using thematic analysis. Quantitative analysis revealed significantly elevated scores for the majority of variables, including empathy and listening, fluency, lexicon, clarity, engagement, functionality, aesthetics, information, and perceived impact. The interview findings demonstrated a notable level of appreciation for the intervention. The participants described the chatbot as engaging and supportive, highlighting a smooth interaction flow, content-appropriate language, and messages of suitable length. The REA prototype demonstrated high acceptability, usability, and perceived usefulness among a diverse range of stakeholders, thus supporting its potential as a scalable, stigma-reducing tool for the prevention of PPD. Subsequent research endeavours will focus on refining the chatbot’s personalisation features and conducting comprehensive clinical trials to evaluate its efficacy.

## 1. Background

Recent evidence highlights an emerging need for psychological support within the general population, particularly among vulnerable groups exposed to elevated levels of stress ([28]; [33]). Pregnancy is a period of heightened vulnerability marked by profound physical, psychological, and social changes. Numerous studies report that pregnant women frequently experience psychological symptoms, with anxiety, stress, and depression among the most common ([2]; [19]).

Postpartum depression (PPD) prevalence varies depending on severity, with mild forms (baby blues) affecting 30–80% of mothers ([26]; [34]), and clinically significant PPD occurring in 10–15% ([23]). Diagnosis and treatment are often hindered by women concealing symptoms, leading to the term “smiling depression” ([8]). Severe cases may present confusion, hallucinations, and delusions, with confusion being particularly persistent ([1]; [4]).

Preventive educational and psychosocial interventions have shown promise ([6]), but few structured programs are widely implemented. Early prevention is essential, as symptoms can impair mother–infant bonding within the first three months postpartum ([3]; [11]).

Psychological interventions during pregnancy are effective ([3]; [5]; [6]; [11]), but access remains limited due to workforce shortages, geographical barriers, and stigma. An Italian survey found that 80% of Mental Health Departments lack structured perinatal mental health pathways, and Mother–Baby Units are absent nationwide ([18]).

The WHO promotes scalable, inclusive mental health strategies, with digital interventions emerging as promising tools to enhance accessibility, flexibility, and reduce stigma ([10]).

### 1.1. Acceptance and Commitment Therapy (ACT) in the Perinatal Period

This study aims to assess the acceptability and feasibility of a digital Acceptance and Commitment Therapy (ACT)–based intervention to promote psychological well-being and prevent perinatal mental disorders, including PPD. ACT emphasises acceptance and psychological flexibility, helping individuals disengage from harmful cognitive fusion and reconnect with personal values ([14], [13]). ACT proposes that attempts to control or avoid unwanted internal experiences—such as thoughts, emotions, or memories—are often counterproductive ([14]). Individuals may become cognitively “fused” with these experiences, interpreting them as literal truths and thereby shifting focus away from personal values and goals. By promoting psychological flexibility and acceptance, ACT seeks to help individuals reconnect with what is most meaningful to them.

In the context of PPD, the intense stress and psychological disruption following childbirth may trigger intrusive or distressing thoughts. While some cognitive fusion may be harmless, fusion with highly disturbing cognitions—such as “I wish I didn’t have this baby”—can have serious consequences. When such thoughts are believed or acted upon, they may lead to feelings of guilt or emotional disconnection from the infant. Furthermore, PPD can involve distorted perceptions, including hallucinations or delusions, which may further undermine mental health and caregiving capacity ([1]).

The content architecture of the digital chatbot intervention was derived from a specialised manual based on ACT to prevent PPD ([17]). Core ACT processes—namely acceptance, cognitive defusion, present-moment awareness, value clarification, and committed action—were extracted from the manual and reformulated into concise, conversational modules suitable for chatbot delivery.

### 1.2. Digital Interventions for Perinatal Mental Health

Recent publications have indicated the potential of telemedicine and digital interventions to reduce perinatal mood disorders effectively. A recent systematic review found that approximately 62% of RCT studies on television interventions (chat, apps, websites) showed a significant improvement in depressive symptoms compared to control groups ([20]). In particular, content derived from digital cognitive behavioural therapy (CBT) and modulated in a preventive context has proven effective, especially when integrated with human or peer support ([20]; [24]).

Additionally, several chatbots have been developed and evaluated, with a specific focus on pregnant women and women in the postpartum period. The “Moment for Parents” study employed a user-centred design, incorporating ethnographic interviews (n = 43) and a pilot project with 108 participants. The results demonstrated above-average levels of engagement and re-engagement, with a resumption of interaction observed in 63.9% of users who had previously ceased using the chatbot. Of these, 40.6% subsequently used the chatbot a minimum of three times. A significant proportion of respondents expressed satisfaction with the content, with 93.3% rating it as relevant ([9]).

In the field of mental health research, there is a clear preference for rule-based models over generative ones. This preference is rooted in the belief that rule-based models offer greater consistency, contextualised empathy, and reliability in responding to users with postpartum mood disorders ([35]).

Recently, a published protocol introduced “ALBA”, a virtual coach chatbot specifically based on ACT techniques to promote quality of life and perinatal psychological well-being ([27]). This constitutes the inaugural documented instance of an ACT-based digital intervention that has been formally developed with perinatal objectives.

Finally, it is essential to emphasise that digital strategies, while promising, require careful design tailored to specific needs (e.g., accessibility, culturally sensitive content) and a clear definition of the target audience (e.g., universal vs. selective prevention) to maximise their effectiveness and clinical adoption ([9]; [20]).

The ACT protocol for postpartum depression was developed by Espen Klausen from M.S., Università del Wisconsin-Milwaukee ([17]). While there is a paucity of studies that provide unequivocal evidence regarding the efficacy of this protocol, it is acknowledged within the scientific literature that ACT is a widely utilised technique for the prevention of DPP ([32]). Moreover, research conducted on the utilisation of web-based interventions for expectant mothers has demonstrated that the provision of psychoeducational support can serve as a preventative measure against postpartum depression ([12]). Drawing upon these findings, it was determined that the implementation of the ACT protocol within a mobile application would be a viable solution, with the objective of ensuring that every pregnant woman in need has access to this intervention. A notable strength of this approach is the utilisation of a virtual coach, which can provide ongoing support and guidance to the user throughout the intervention.

### 1.3. Present Research

The intervention’s objective is to enhance psychological flexibility in women through exercises based on the ACT model, to prevent postpartum distress.

Based on ACT rationale and evidence on perinatal digital interventions, we have adapted ACT content into short, progressive conversational modules delivered by REA (virtual coach). The aim is to prevent perinatal distress by providing practical exercises, reinforcement and psychoeducation in a flexible, accessible way that can potentially be integrated into care pathways.

The content is delivered through a meticulously designed step-by-step process, in order to promote self-awareness, acceptance, and normalisation of internal states, stress management, and an overall improvement in psychological well-being. This ACT intervention will be fully available to users through digital tools. In particular, it will be delivered by a mobile application and guided by a virtual assistant, REA. The present research aims to assess the prototype of the REA application, gathering feedback and needs from key stakeholders to further refine the application from the perspectives of usability, accessibility, and acceptability of the intervention delivered via chatbot.

The following essay will provide a comprehensive overview of the relevant literature on the subject.

## 2. Materials and Methods

The ACT intervention program is developed iteratively, following the ORBIT model ([7]), which depicts the pathway for translating a human-guided intervention into a potential Digital Therapeutics (DTx).

The multidisciplinary project team, comprising experts in psychology, eHealth research, and communication, held biweekly meetings during the design and development phase. A user-centred design approach ensured continuous involvement of patients, healthcare providers, and security specialists. Development followed an iterative process through (i) intervention content development (identified and adjusted from the ACT protocol) and (ii) iterative software development of a prototype and formative evaluation. The development and iterative processes of the ACT-based intervention are shown in Figure 1.

### 2.1. Intervention Content Development

Adapting the protocol into a digital format began with a comprehensive review of both the literature and the original manual. Once the week-by-week thematic structure was defined, a team of psychologists with expertise in communication adapted the material into a chatbot-based, individualised digital format. The process started with the development of the chatbot’s dialogue scripts, followed by the creation of multimedia resources (videos, audio, and images) to convey key information in a format more accessible and engaging for women than text alone.

The initial intervention content was drafted by the principal investigator and subsequently refined by the entire research team through iterative cycles, resulting in an eight-session program delivered electronically through a combination of text, audio, video, and images. User testing was conducted to ensure accessibility, compatibility with the app format, and preservation of the intervention’s scientific integrity. Table 1 outlines the eight sessions and their respective topics.

The intervention was designed so that REA—the chatbot—always initiates the interaction. At this stage, users cannot submit specific queries; instead, the chatbot delivers information and prompts self-reflection through targeted questions. User responses are not analysed in the current phase, as this lies outside the scope of the study.

The intervention is scheduled to span eight weeks, with six weeks occurring before delivery and the subsequent two weeks following delivery. Each session lasts approximately 20 min.

The app is designed to include four sections: Chatbot, Diary, Gallery, and Progress. The first low-fidelity prototype version of the app was developed (Figure 2).

#### Functions of App Sections

The primary function of REA is to intervene to prevent the onset of disease and to address the issue of postpartum depression. The following characteristics are of particular significance.

During the intervention, the user is guided by a chatbot, REA, in completing eight stages over eight weeks (six before delivery and two after).

On the designated day, the participant is provided with psychoeducational material by REA, after which they undertake exercises during the week and record their progress in a diary.

It is also possible for the user to consult all the material displayed by REA in the Gallery. As time progresses, a tree will be observed growing in the Path section, thereby symbolising the progression and development of the intervention.

The intervention is accessed from the application via a dedicated button. Initially, the chatbot materialises. Thereafter, the user can navigate and explore the various sections of the application. The following essay will provide a comprehensive overview of the relevant literature on the subject.

The main menu presents the user with all the sections of the application. The “Exercises” section documents all the proposed exercises from the sessions. Each card comprises an illustration, a title, and a concise explanation. The weeks are displayed in descending order. The “Diary” function provides a designated space where users can articulate their reflections, sentiments, and experiences in the period between interactions with the chatbot. Within the gallery, users can view and download all multimedia content to their mobile phones, categorised by week, including all videos, audio files, and images provided by REA.

The utilisation of the button located adjacent to the search bar enables the user to implement a content filter. The ‘path’ section provides a visual representation of the user’s progress through the intervention, organised into a hierarchical structure. At the conclusion of the week, and thus of the session, the tree is observed to be growing. From this location, it is also feasible to observe keywords associated with the present session.

### 2.2. Iterative Software Development and Formative Evaluation

Adapting ACT interventions through the implementation of the REA chatbot offers a novel approach to delivering psychological support. Users can participate in comprehensive and effective interventions through personalised sessions. In the initial session, users are prompted to choose one day of the week on which they wish to be contacted by REA, along with their preferred time slot (morning, afternoon, or evening). Based on user responses, REA provides tailored content—for example, REA asks whether the user would like to review the previous session, whether they prefer to complete a specific exercise immediately or later, and so on, with the dialogue branching accordingly.

In the first iteration, conducted in January 2025, two psychologists and two communication experts tested a preliminary prototype, providing feedback to ensure the intervention program was logically structured. At this co-design stage, the involvement of experienced professionals, alongside end users, was considered crucial. For this reason, in a subsequent study (Test 1), four additional psychologists and four communication experts were involved to test the low-fidelity prototype that was created. Later, end users were also engaged in the process (Test 2), including seven women and four clinicians.

Overall, a multistage evaluation of the REA prototype was carried out between May and July 2025, involving a total of 19 participants, as illustrated in Figure 3.

#### 2.2.1. Study Aims: Quantitative and Qualitative Components

Using a mixed methods approach, the study aims first and foremost to quantitatively verify the usability, perceived quality and potential impact of the ACT-based digital intervention delivered via chatbot. To this end, user evaluations are collected using standardised 5-point scale tools and compared with a predefined benchmark equal to the midpoint of the scale (3/5), interpreted as the threshold of acceptability. The objective is to establish whether the median values exceed this threshold, indicating a level of user experience and perceived value that is at least ‘good’ at this prototypical stage. The choice of a comparison with the midpoint reflects the formative nature of the study and allows practical insights to be drawn for design iterations.

In parallel, the qualitative component delves deeper into the user experience to understand how people perceive the tone and clarity of communication of the chatbot, the structure and rhythm of the modules, the relevance of the content to the perinatal period, and the degree of personalisation perceived. Needs, barriers, and concrete suggestions for improvement are explored.

#### 2.2.2. Variables Identification

To collect the necessary data, key variables were identified for evaluation: communication, session structure, materials, content, engagement, functionality, aesthetics, information, subjective quality, and perceived impact. The first four variables were assessed using the Semantic Differential tool ([25]), and the subsequent six with the User Version of the Mobile Application Rating Scale (uMARS) ([29]). Following completion of the questionnaires, an ad hoc semi-structured interview was conducted to explore selected variables in greater depth.

The Semantic Differential is a measurement tool consisting of a series of scales, each comprising a pair of bipolar adjectives anchored by a five-point rating continuum. Based on the study’s focus variables, a set of sub-variables was defined, and ad hoc items were developed for this research. Appendix A Table A1 presents the selected variables, along with their respective sub-variables and corresponding items.

The uMARS questionnaire assesses mobile applications across four objective dimensions—engagement, functionality, aesthetics, and information—and one subjective dimension. It comprises 20 items distributed as follows: engagement (n = 5), functionality (n = 4), aesthetics (n = 3), and information (n = 4), plus 4 items assessing subjective quality. An additional section on perceived impact (6 items) evaluates users’ perceptions of the app’s overall usefulness. Responses are recorded on a 5-point scale (1 = inadequate, 2 = poor, 3 = acceptable, 4 = good, 5 = excellent), providing a standardised measure of the usability and quality of mobile health applications.

Interviews, by contrast, allow for a more in-depth exploration of users’ attitudes and preferences toward new technological solutions. Open-ended discussions can provide researchers with richer insights into potential issues and concerns related to the future adoption of such innovations ([21]). Appendix B Table A2 presents the list of topics and guiding questions used during the interviews.

#### 2.2.3. Procedure

All procedures were conducted confidentially and only after obtaining written informed consent from all participants. Recruitment relied on word-of-mouth and personal networks, accompanied by a transparent and detailed explanation of the study’s aims. Eligibility was restricted to volunteers aged 18 years or older. Before any study activity, prospective participants received via email an information sheet and a consent form covering participation and data handling, and provided consent electronically.

Operationally, the study proceeded as follows:(1)Presentation of the participant information sheet and informed consent;(2)A two-week interaction period with the REA prototype (one session every other day), with each session lasting about 15 min;(3)Testing on the participant’s own smartphone;(4)Exploration of the interface and sections through mock-ups;(5)ACT-based dialogue sessions delivered via WhatsApp, using structured prompts with predefined buttons and optional free-text responses;(6)Free-text inputs were not analysed at this prototype stage, as this was outside the primary objectives of the study;(7)Participants were informed that reminders and activity feedback would be implemented in future versions.

At the end of the interaction period, participants evaluated their experience by completing two questionnaires: the Semantic Differential instrument and the Italian User version of uMARS ([22]).

A brief semi-structured interview was then conducted online (lasting approximately 20 min). The interview guide was developed specifically for this study to elicit expectations, preferences, and concerns regarding the tested REA solution. Inclusion criteria for the interview were completion of the prototype testing and both questionnaires, as well as consent for the additional interview phase; exclusion criteria were dropout during testing, incomplete questionnaires, or refusal to be interviewed. A total of 10 interviews were carried out by the researcher, audio-recorded to support in-depth analysis, and subsequently examined using qualitative methods. Each interview began with a brief overview of objectives, followed by semi-structured questions about expected use and preferences.

All materials were collected in Italian and pseudonymised; interview audio recordings were stored confidentially and analysed with subjects identified solely by numeric codes. At study completion, participants could request a summary report of the results from the study lead.

In line with Italian Law No. 3/2018, ethics committee approval was not required because the work concerned the usability assessment of a prototype digital intervention, did not involve patients or vulnerable populations (participants included mindfulness experts, psychologists, midwives, and women with relevant experience), and did not include clinical treatment or the collection of sensitive health data. Nonetheless, core ethical safeguards were upheld, including voluntary informed consent and data privacy protection.

No compensation was provided to participants.

#### 2.2.4. Data Analysis

Quantitative data from the Semantic Differential and uMARS questionnaires were analysed using Jamovi, version 2.3.28.0 ([15]). Given the ordinal nature of the measures and the small sample size, we used the Wilcoxon signed-rank test (W) one-sample as a non-parametric alternative to the one-sample *t*-test. ([16]; [31]). The reference value μ = 3 was defined a priori because it corresponds to the midpoint of the 5-point scales used: for uMARS, the value 3 represents the ‘acceptable’ anchor, while for the Semantic Differential, it is the midpoint of the bipolar scale. This benchmark allows us to verify whether the medians of the evaluations deviate significantly (and in practice exceed) a level of sufficiency/acceptability. Rank-biserial correlation (r) and corresponding 95% confidence intervals (CI) were reported to indicate effect size ([31]). All analysis results were considered significant with a critical *p*-value set at 0.05.

Qualitative data from the interviews were analysed thematically ([30]). First, all transcripts were reviewed to gain an overall understanding. Thematic analysis was then conducted by organising themes into tables according to context and participant type (e.g., psychologists, clinicians). Responses were grouped into sub-variables derived from the initial ad hoc topics around which the interview questions were formulated, to address thematic redundancy within the sample. Interviews were conducted and recorded by one author; transcription and analysis were carried out independently by two other authors. Thematic relevance and redundancy were determined by consensus; disagreements were resolved through consultation with a third author, applying a two-out-of-three agreement rule. A final report was produced summarising the main findings, with references to specific participant groups where applicable.

## 3. Results

The pilot evaluation was conducted with a total of 19 participants, comprising four psychologists, four communication experts, seven mothers, and three midwives. Of these, 16 responded to the questionnaire (three psychologists, four communication experts, five mothers and four midwives), and 10 also participated in the semi-structured interview (three psychologists, three communication experts, two mothers and two midwives).

The mean age of the subjects who completed the questionnaire was 34.5 years (SD = 7.89, ranging from a minimum of 26 to a maximum of 55 years). The mean number of years of schooling for the sample was 17.88 (SD 2.94, range 13–22 years of education).

### 3.1. Quantitative Results

The utilisation of a semantic differential-based questionnaire enables the observation of the respondents’ mean positioning with respect to the four macro-variables under investigation. Specifically, with regard to the sub-variables, the Wilcoxon analysis yielded several significant results, as detailed in Table 2. The graphical representation of mean values derived from the Semantic Differential for each item is noted in Appendix A Table A1. As demonstrated in Table A1 in the Appendix A, a propensity towards the positive semantic pole (right pole) is discernible in the feedback from these groups of participants.

The analysis was conducted using a one-sample Wilcoxon signed-rank test, with a reference value of μ = 3. Results showed statistically significant differences for almost all evaluated dimensions, with the sole exception of Session Duration.

The results highlight that users’ perceptions of several aspects of the experience are highly positive. Elements related to communication quality (empathy, fluency, lexicon, clarity), multimedia richness (audio tracks, images, and videos), and overall content evaluation were rated substantially above the reference mean. The effect sizes, approaching the maximum possible value, indicate an almost unanimous consensus in a positive direction.

The only indicator that did not differ significantly from the reference value was Interaction Duration, suggesting that session length was not perceived as notably different from the expected level. This finding can be interpreted as a sign of balance: the experience is considered satisfying without being perceived as excessively long or short.

The uMARS evaluated the respondents’ average positioning in four key dimensions. Detailed results from the Wilcoxon tests are summarised in Table 3.

A one-sample Wilcoxon signed-rank test was conducted with a reference value of μ = 3. Results indicated statistically significant differences for most evaluated dimensions.

Specifically, Engagement (W = 108.00, *p* = 0.007, r = 0.80), Functionality (W = 121.00, *p* = 0.006, r = 0.78), Aesthetics (W = 105.00, *p* = 0.011, r = 0.75), Information (W = 120.00, *p* < 0.001, r = 1.00), and Perceived Impact (W = 79.00, *p* = 0.020, r = 0.73) all scored significantly higher than the reference value, with large effect sizes.

In contrast, Subjective items (W = 38.00, *p* = 0.304, r = 0.38) did not show a significant difference, and the effect size was small-to-moderate.

The findings suggest that users evaluated several aspects of the experience very positively. Notably, factors such as engagement, functionality, aesthetics, information quality, and perceived impact received scores significantly above the reference mean, accompanied by large effect sizes, indicating a strong and consistent preference in a positive direction.

Conversely, subjective quality did not differ significantly from the benchmark value, suggesting that while participants appreciated specific functional and aesthetic features, their overall subjective assessment of quality may not have been distinctly higher than average. This could imply that although the product/service excels in certain tangible dimensions, perceptions of its overall quality may be more nuanced or influenced by additional factors not measured in this test.

### 3.2. Qualitative Results

Two psychologists conducted the qualitative interviews to gather additional information. Interviews were conducted and analysed according to the identified thematic variables.

The thematic analysis produced four main themes with related sub-themes: (1) communication and interaction with the chatbot (tone and perceived empathy; fluidity of messages; variety of response options; balance between closed and open questions; clarity of vocabulary); (2) module structure (session duration; message length; perception of repetition); (3) multimedia materials (adequacy of audio/video; perception of ‘coldness’ vs. realism; consistency with adult target audience); (4) content and personalisation (relevance for pregnant women; depth of responses; inclusiveness for vulnerable groups; references to previous sessions).

#### 3.2.1. Theme 1: Communication and Interaction with the Chatbot

The chatbot received positive evaluations from the vast majority of experts. It was described as patient, welcoming and engaging—qualities that contributed to enhancing user involvement. One communication expert described its communicative style as simultaneously formal and friendly, while a psychologist referred to the chatbot as “clumsy yet polite.”

The overall interaction was generally well received by representatives from all four expert categories, who described it as easy, quick, and undemanding, noting also that the chatbot effectively took into account the users’ responses. The reminders provided to participants throughout the week were generally appreciated. One psychologist emphasised the value of being able to pause the interaction.

Nonetheless, one mother pointed out the limitations of artificial intelligence, noting that the chatbot appeared unable to fully comprehend users’ thoughts and emotions, and that the interaction was perceived as overly guided and rigid—a sentiment shared by a usability expert.

A midwife reported that the chatbot occasionally sent multiple messages in quick succession, which made reading difficult, as she frequently had to scroll back through the conversation.

A psychologist mentioned losing track of the dialogue due to difficulty in recalling his own thoughts and reflections during the exchange.

Those who tested the application valued the inclusion of both open-ended questions, which allowed users to answer freely in their own words, and closed-ended questions.

While the number of pre-defined responses in the closed-ended questions was generally considered adequate, some usability experts expressed a preference for a greater variety of response options and a reduction in the number of such questions overall. Additionally, there was a preference against the use of pre-defined replies, considered less appealing, and the excessive use of closed-ended questions was seen as contributing to a tedious user experience.

Regarding the vocabulary employed, it was generally perceived as clear and understandable. However, one usability expert raised concerns that some of the terminology used was too complex and more suited to a highly educated population.

Finally, one mother found it somewhat difficult to respond to abstract prompts, such as those asking what thoughts, images, or emotions come to mind when reading words like “milk” or “parenting.”

#### 3.2.2. Theme 2: Module Structure

A psychologist appreciated the duration of each session, considering it appropriate and not excessively long.

Although there was a shared perception that the sentence lengths were adequate, another psychologist noted that some messages were rather too long. Finally, a mother observed that the repetition of some exercises across different sessions disrupted the overall flow of the interaction.

#### 3.2.3. Theme 3: Multimedia Materials

While one mother appreciated the simplicity and clarity of the videos and animations presented during the intervention, which focused on key concepts and vocabulary, another perceived the audio-visual content as emotionally distant and lacking empathy.

Two usability experts expressed concerns about the repetition of materials across multiple sessions, noting that it could feel repetitive and potentially discourage users from continuing to use the app.

Additionally, a psychologist found the videos to be excessively minimalistic and lacking in realism.

#### 3.2.4. Theme 4: Content and Personalization

The content presented was positively evaluated by midwives, who considered it suitable for pregnant women. In particular, the language was perceived as clear and accessible, yet neither trivial nor simplistic, and engaging. Moreover, referencing content from previous sessions was appreciated.

A psychologist highlighted the clarity of the information, referring not only to the content presented in the videos but also to that conveyed directly by the chatbot.

However, one midwife expressed concerns regarding the suitability of the content for disadvantaged groups (e.g., immigrant women and women with limited formal education).

Additionally, a mother and a usability expert would have preferred more detailed responses, while another usability expert felt that the content was redundant across sessions. Psychologists also noted that the content lacked personalisation, with insufficient attention paid to the users’ individual thoughts.

#### 3.2.5. Overall Assessment

The participants reported an average rating of 8.3 on a scale of 1 to 10, indicating a generally positive overall evaluation of the intervention. The lowest score recorded was 7.5, reported by a psychologist, while the highest—10—was reported by a mother.

When asked about the most suitable time for delivering the intervention, most participants identified the second half of pregnancy as the most appropriate period. However, some psychologists and usability experts emphasised the preventative value of introducing the intervention earlier (e.g., after the fourth month), to provide strategies that could be useful both during pregnancy and in the postnatal period. A mother suggested that the intervention could be delivered in parallel with antenatal education courses.

#### 3.2.6. Insights into the Effectiveness of the Intervention

Regarding aspects of effectiveness, a midwife emphasised that for the intervention to be truly effective, it must be supported by a structured programme involving a professional. Similarly, a usability expert regarded it as a supplementary tool.

Additionally, one mother expressed that this intervention could be particularly beneficial for women experiencing pregnancy for the first time.

Table 4 summarises the main positive and negative aspects identified in the user interviews with supporting quotes.

#### 3.2.7. Suggestions for Improvement

During the semi-structured interviews, participants were free to share suggestions, from which the following key points emerged:Enhance the realism of video content—a psychologist noted that, although they found the current material appealing, incorporating more realistic videos could better engage adult women, as the current videos appear to be primarily targeted at children and adolescents.Clarify the chatbot’s purpose and limitations. According to a psychologist, it is essential to manage users’ expectations to avoid unrealistic assumptions and to ensure they understand that guidance will be provided throughout the intervention.Users could be encouraged to record their thoughts and reflections throughout the intervention—as noted by a psychologist, this may help them maintain coherence in their thinking and clarify the underlying psychological constructs.Add a section dedicated to supplementary material—a psychologist highlighted the potential benefit of offering additional resources or external links for users interested in deepening their understanding of specific concepts.Enhance initial rapport-building and space to express themselves and feel more actively involved in the interaction—while a psychologist proposed engaging the user by building a connection at the beginning of the intervention through personal questions (e.g., ‘What is your occupation?’), a mother suggested that the chatbot could increase user involvement by asking questions on a weekly basis.Reduce repetitiveness in message content—a communication expert expressed concern about the repetition of certain messages across sessions, suggesting that key concepts could be reiterated using varied wording and phrasing.

## 4. Discussion

The present study investigated the adaptation of Acceptance and Commitment Therapy (ACT) techniques for the prevention of postpartum distress and depression through REA, a virtual coach embedded within a mobile application. The programme is scheduled to span a period of eight weeks, with six sessions to be delivered antenatally and two sessions to be delivered postnatally. Each session integrates interactive exercises, multimedia content, and structured guidance, with the objective of providing support during the period of pregnancy and early motherhood characterised by significant emotional intensity.

The ORBIT framework was employed to meticulously refine the dialogues and interface prior to pilot testing. The participants, who included psychologists, midwives, communication experts and mothers, reported an overall positive experience. Quantitative data from the Semantic Differential and uMARS scales indicated high satisfaction across most dimensions, particularly in communication quality, content clarity, multimedia richness, engagement, functionality, and perceived impact. The magnitude of the effect sizes was consistently substantial, indicating a predominant and nearly unanimous positive perception among users.

The findings were complemented by qualitative feedback. It has been posited that REA exhibited qualities of patience, a welcoming demeanour, and an engaging presence. The hybrid interaction format, which incorporated both open- and closed-ended questions, was met with approval by the participants. The chatbot’s interaction flow was described as smooth, and it was rated as easy to use by the participants. Furthermore, the reminders designed to encourage sustained engagement were well-received. Midwives expressed satisfaction with the content, citing its relevance and appropriateness for expectant mothers. The chatbot was also perceived as using terminology appropriate to the content, while maintaining a suitable length of messages and sentences. Conversely, mothers emphasised the programme’s potential benefits during the latter stages of pregnancy and for novice parents.

### Strengths, Limitations, and Future Directions

The findings of this study suggest that the utilisation of chatbots for the delivery of ACT interventions has the potential to offer significant psychological support and guidance to pregnant women. The structured format ensures fidelity to evidence-based content, while the digital delivery facilitates broad accessibility and scalability. Participants noted that, while the majority of aspects were highly satisfactory, limitations emerged with regard to personalisation, repetitiveness and the emotional depth of interactions. This finding indicates that even well-designed digital interventions may benefit from additional features that enhance the experience, rendering it more adaptive and tailored to individual needs. In line with ACT principles, the intervention targets psychological flexibility through brief, values-oriented exercises adapted to the perinatal context. To address these gaps, next iterations will prioritise adaptive branching, context-aware tailoring (e.g., trimester, parity, risk status), and richer affective cues to deepen the therapeutic alliance.

The finding that the second half of pregnancy was considered the most suitable period for engagement is in alignment with previous research emphasising the importance of timing in perinatal interventions. Furthermore, participants proposed that the integration of digital support with professional guidance could enhance effectiveness, particularly for first-time mothers or those at higher risk of postpartum distress. Future designs could test trimester-specific entry points and session pacing windows, including postpartum boosters, to optimise timing. We also envisage a stepped-care model that integrates midwives and mental health professionals for triage and escalation when risk flags are detected.

The present findings are consistent with the extant evidence that digital mental health tools have the capacity to promote engagement, provide psychoeducational support, and encourage positive behaviour change during pregnancy. As with other studies in this field, the participants expressed appreciation for content that is clear and structured, and for user-friendly interfaces. Concurrently, challenges in replicating human-like empathy, responsiveness, and personalisation, as previously identified in research on chatbots, persist. Feedback regarding repetitive content and limited flexibility reflects ongoing difficulties in designing fully adaptive automated interventions, highlighting areas for further improvement.

A significant strength of this study is its participatory development process. The involvement of both professional experts and pregnant women in the design and evaluation process was instrumental in ensuring the relevance, clarity and usability of the intervention. The integration of quantitative and qualitative methodologies yielded complementary insights into user experience and perceived value.

Nevertheless, the study is not without its limitations. The small, highly educated sample may limit the generalisability of the results, and the evaluation did not include clinical outcomes, such as reductions in stress or depressive symptoms. Furthermore, the relatively limited timeframe and close session frequency may have contributed to perceptions of repetition. Further research is required to examine long-term engagement and effectiveness in diverse real-world settings. The sample was non-random and there was no control or comparison group, increasing the risk of selection and expectancy biases. Measures were self-reported, the prototype was low-fidelity, and novelty or demand characteristics may have influenced responses.

Subsequent iterations of the software should concentrate on enhancing personalisation, increasing flexibility, and improving the realism of multimedia content. One potential solution to this issue could be to adapt the session frequency or to allow more user-driven pacing, as this could potentially reduce perceived repetition and thereby sustain engagement. It is imperative that the intervention is tested in larger, more diverse populations, including women with lower educational attainment or from minority backgrounds, in order to assess its accessibility and inclusivity. Integration with professional antenatal care or educational programmes has the potential to enhance both impact and user trust. The next phase will prioritise recruitment of pregnant women for in situ testing, with adequate power and longitudinal follow-up to assess usability, acceptability, and preliminary clinical signals. Planned outcomes include psychological flexibility as a process measure and validated clinical endpoints, alongside implementation metrics such as feasibility, acceptability, and cost-effectiveness.

## 5. Conclusions

The present study evaluated an ACT-based intervention delivered via the REA chatbot, which was designed to prevent postpartum distress and support maternal well-being. The protocol, adapted for pregnant women, incorporated interactive exercises, gamification, and personalised feedback to enhance engagement. Utilising the ORBIT framework, the validation of the chatbot’s structured dialogues was conducted, thereby confirming the effective delivery of content while concomitantly identifying areas that would benefit from enhancement.

Participants identified the app’s clear organisation, intuitive interface, session pacing, animations, and the emphasis on revisiting core content as key strengths. Concurrently, limitations were identified, including restricted personalisation, intermittent content repetition, and terminology that may present challenges for some users. It was posited by respondents that combining the digital intervention with structured in-person support could further enhance its impact, particularly for first-time mothers or higher-risk populations.

Future development should concentrate on enhancing personalisation, engagement, and accessibility, in addition to conducting further usability testing in a range of healthcare settings. The findings of this study indicate that digital interventions such as REA offer a scalable, accessible, and evidence-informed approach to supporting maternal mental health during pregnancy and the postpartum period.

## Figures and Tables

**Figure 1 behavsci-15-01723-f001:**
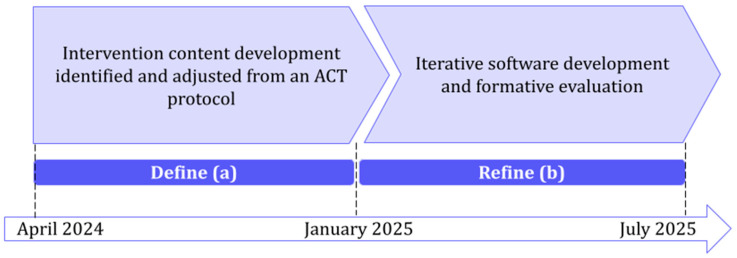
Development and iterative processes of the ACT-based intervention (Phase 1 of the ORBIT model—Design).

**Figure 2 behavsci-15-01723-f002:**
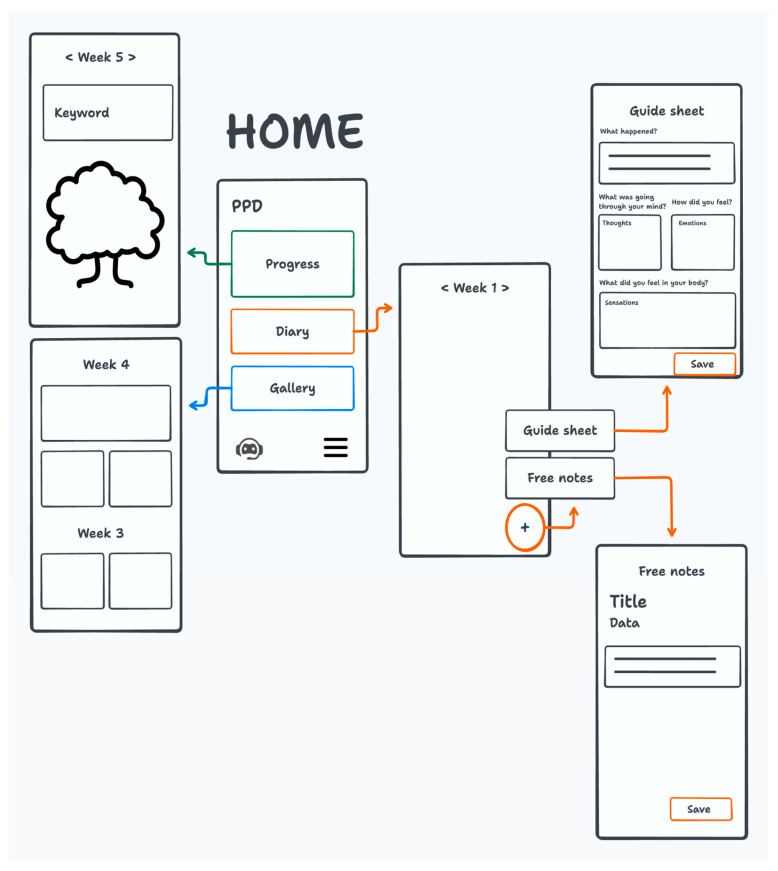
Low-fidelity prototype version of the REA app.

**Figure 3 behavsci-15-01723-f003:**
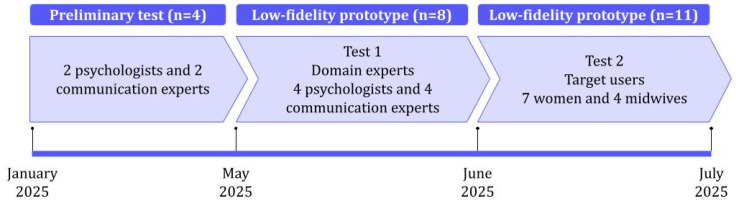
Timeline of software development and formative evaluation specifying the people involved (N = 23).

**Table 1 behavsci-15-01723-t001:** Description of the topics covered week by week.

Week	Topics
Week 0	Screening and Enrolment.
Week 1	Awareness, values, automatic and conscious thoughts
Week 2	Practice, contacting the present moment
Week 3	Acceptance, adjusting thoughts
Week 4	Committed action, setting priorities
Week 5	Defusion, detachment from thoughts
Week 6	Focusing, different facets of the self
Week 7	Contacting the present moment, review
Week 8	Acceptance, conclusions

**Table 2 behavsci-15-01723-t002:** Results from the Semantic Differential (N = 16).

	Participants, n (%)	Values, Mean (SD)	Values, Median (Range)	*W*	*p* Value	r
Empathy and listening	16 (100)	3.76 (0.53)	3.70 (3.00–5.00)	120.00	<0.001	1.00
Smoothness and fluidity	16 (100)	3.94 (0.57)	4.00 (3.00–5.00)	91.00	<0.001	1.00
Chatbot interaction	16 (100)	3.72 (0.56)	3.75 (2.25–4.75)	114.00	0.002	0.90
Lexicon	16 (100)	4.34 (0.81)	4.50 (2.00–5.00)	131.50	<0.001	0.93
Session duration	16 (100)	3.00 (0.66)	3.00 (2.00–4.50)	21.00	0.904	−0.07
Audio tracks	16 (100)	3.78 (0.50)	4.00 (3.00–5.00)	120.00	<0.001	1.00
Infographics and videos	16 (100)	3.81 (0.76)	4.00 (2.34–5.00)	101.00	0.002	0.92
Content evaluation	16 (100)	3.98 (0.59)	4.00 (2.67–5.00)	119.00	<0.001	0.98
Content clarity	16 (100)	4.44 (0.63)	4.75 (3.50–5.00)	136.00	<0.001	1.00

Note. H_a_ μ ≠ 3.

**Table 3 behavsci-15-01723-t003:** Results from the uMARS (N = 16).

	Participants, n (%)	Values, Mean (SD)	Values, Median (Range)	*W*	*p* Value	r
Engagement	16 (100)	3.45 (0.51)	3.40 (2.40–4.40)	108.00	0.007	0.80
Functionality	16 (100)	4.13 (0.93)	4.25 (1.00–5.00)	121.00	0.006	0.78
Aesthetics	16 (100)	3.79 (0.89)	4.00 (1.00–4.67)	105.00	0.011	0.75
Information	16 (100)	4.20 (0.53)	4.25 (3.00–5.00)	120.00	<0.001	1.00
Subjective items	16 (100)	3.17 (0.54)	3.00 (2.50–4.50)	38.00	0.304	0.38
Perceived impact	16 (100)	3.56 (0.72)	3.67 (1.67–5.00)	79.00	0.020	0.74

Note. H_a_ μ ≠ 3.

**Table 4 behavsci-15-01723-t004:** Main positive and negative aspects emerged from user interviews.

Variable and Subvariable	Positive Aspects	Negative Aspects	Quotes
**Communication**			
Empathy & Listening	Engaging (n = 4), encouraging (n = 4), patient, welcoming, and friendly.	Lack of recognition of emotions and feelings.	“I appreciated how it managed to be both formal and friendly”.
Flow & Smoothness	Hybrid interaction (n = 2), including both open-ended and closed-ended questions. Smooth interaction flow (n = 3) and effective communication style.	Delays between consecutively generated messages by the chatbot, difficulties in expressing and tracking one’s own thoughts, and challenges in formulating responses to certain questions.	“I found it a bit difficult when I was asked to express my thoughts after being given the words ‘milk’ or ‘parenting.’”
Chatbot Interaction	Adequate number of response options (n = 3), ease of use (n = 4), comfort, interactivity (n = 2), and overall pleasant experience. The use of reminders (n = 2) and the option to pause were appreciated.	Inadequate response options, a predominance of closed-ended questions, rigidity, boredom, limited interactivity, and low personalisation.	“The chatbot allowed you to choose whether to continue or not; you weren’t obliged to, so in my opinion, that’s fair.”
Lexical Choice	Accessible language (n = 2), and terminology appropriate to the content (n = 6).	Terminology suited mainly to a highly educated population.	“The language used appeared overly technical for a general audience, […] some terms and expressions were, in my opinion, not easily understandable.”
**Module Structure**			
Single Interaction Duration	Appropriate length of messages and sentences (n = 5), which are associated with videos and exercises. Brief sessions.	Redundancy across sessions, excessive message length.	“I found it positive that the sessions were fairly short.”
**Materials**			
Audio Tracks	__^a^	Coldness and low empathy.	“The videos felt really cold to me, meaning they lacked empathy.”
Images & Videos	Simple animations, clear concepts, and emphasis on key words.	Repetition (n = 2). Videos were overly minimal and lacked realism.	“The repetitions felt somewhat redundant and, as a result, tedious and boring […].’”
**Content**			
Content Evaluation	Non-trivial language, appropriate for a pregnant audience. Concrete and relevant content.	Surface-level and repetitive content. Shallow and insufficiently personalised responses (n = 2).	“The app is a very useful tool, but only in the hands of people who already have certain kinds of resources”
Content Clarity	Good referencing of previously presented content. Simple and clear.	Not adequately tailored to disadvantaged or vulnerable populations.	“At certain points […], it would provide a recap, which was very helpful for staying engaged with the topic.”
**Insights into the Effectiveness of the Intervention**	__ ^a^	__ ^a^	“It is especially useful for a woman who is pregnant for the first time.”

Note. Numbers in brackets indicate the frequency of responses. ^a^ No statements.

## Data Availability

The datasets used and/or analysed during the current study are available from the corresponding author on reasonable request.

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
