# Peer review of "Prevention of Postpartum Depression via a Digital ACT-Based Intervention: Evaluation of a Prototype Using Multiple Case Studies"

_behavsci, 2025, doi:10.3390/bs15121723_

Round 1

Reviewer 1 Report

Comments and Suggestions for Authors

This article presents the usability evaluation of a prototype chatbot application (REA) designed based on Acceptance and Commitment Therapy (ACT) principles, aiming to support the prevention of Depression During Pregnancy (DPP) in women during late pregnancy and the early postnatal period.        

The study is introduced with appropriate background information. Although several references are quite dated, their inclusion may be justified because they provide foundational contributions to key clinical concepts.

The research methodology is executed appropriately, using a multidisciplinary team and relying on an iterative development of the app prototype following the User-Centred Design (UCD) paradigm.

The analysis of the research findings is conducted effectively using both quantitative methods (two surveys) and qualitative methods (semi-structured interviews). The results obtained are methodologically validated and presented clearly and compellingly.

Finally, the conclusions are consistent with the results obtained and are convincing.

Minor correction: in section 2.1.1. Functions of App sections, lines 203 to 208 should be removed as they are repeated.

Author Response

Thank you very much for taking the time to review this manuscript. Please find the detailed responses below and the corresponding revisions/corrections in red in the re-submitted files.

COMMENT 1 [Minor correction: in section 2.1.1. Functions of App sections, lines 203 to 208 should be removed as they are repeated.]

RESPONSE 1 [We have removed the duplicate sentences in Section 2.1.1 (“Functions of App sections”)]

Reviewer 2 Report

Comments and Suggestions for Authors

1 In the “Background” part, it is essential to introduce the “Acceptance and Commitment Therapy (ACT)” and digital interventions separately.

2 The purpose of the quantitative and qualitative parts of this study can be describe more clearly.

3 Line 369-370, please explain why “The analysis was conducted using a one-sample Wilcoxon signed-rank test, with a 369 reference value of μ = 3.”

4 It may be more beneficial to recruit some prenatal women to use the app and experience the ACT intervention.

5 In the result part, the themes of the qualitative results need to be presented more clearly.

Comments on the Quality of English Language

The paragraphs of the manuscript can be refined more clearly.

Author Response

Thank you very much for taking the time to review this manuscript. Please find the detailed responses below and the corresponding revisions/corrections in red in the re-submitted files.

COMMENT 1 [In the “Background” part, it is essential to introduce the “Acceptance and Commitment Therapy (ACT)” and digital interventions separately.]

RESPONSE 1 [We have restructured the Background section, introducing two separate subsections: 'Acceptance and Commitment Therapy (ACT) in the perinatal period' and 'Digital interventions for perinatal mental health', followed by a brief Rationale on the integration of ACT in the REA chatbot, in the ‘Present research’ subsection.]

COMMENT 2 [The purpose of the quantitative and qualitative parts of this study can be describe more clearly.]

RESPONSE 2 [In the Methods section, we have explained the specific objectives of the quantitative and qualitative components and the rationale behind them in a new subsection named ‘Study aims: quantitative and qualitative components’.]

COMMENT 3 [Line 369-370, please explain why “The analysis was conducted using a one-sample Wilcoxon signed-rank test, with a 369 reference value of μ = 3.”]

RESPONSE 3 [We added an explicit justification in the new subsection “2.2.1. Study aims” and in the subsection “2.2.4 Data analysis”.]

COMMENT 4 [It may be more beneficial to recruit some prenatal women to use the app and experience the ACT intervention.]

RESPONSE 4 [We have expanded subsection 4.1 in the manuscript (Strengths, Limitations, and Future Directions) with a commitment to conduct the next phase with pregnant women (prenatal) in a clinical/territorial setting, subject to ethical approval, to evaluate usability, acceptability and preliminary outcomes.]

COMMENT 5 [In the result part, the themes of the qualitative results need to be presented more clearly.]

RESPONSE 5 [We have reorganised the qualitative results by theme and sub-theme with clear headings and added a summary paragraph at the beginning of the section 3.2.]

Reviewer 3 Report

Comments and Suggestions for Authors

The article is an interesting exploratory work. The number of participants (19) is very small and so conclusions should be out into the context of not solid evidence. Its merit is the exploratory of the work undertaken,

Future research should be recommended in the final section of the article, namely other studies with larger number of participants and statistically sound samples, which is not the case with this study. Please a add a clear section on "Limitations and Further Research" at the end of the Manuscript text.

Author Response

Thank you very much for taking the time to review this manuscript. Please find the detailed responses below and the corresponding revisions/corrections in red in the re-submitted files.

COMMENT 1 [Future research should be recommended in the final section of the article, namely other studies with larger number of participants and statistically sound samples, which is not the case with this study. Please a add a clear section on "Limitations and Further Research" at the end of the Manuscript text.]

RESPONSE 1 [We expanded section 4.1 Strengths, Limitations, and Future Directions, separate from the Discussion, with limitations (small, educated sample; absence of clinical outcomes; reduced algorithmic personalisation; possible repetition effects) and future research directions (study with pregnant women; personalisation; clinical trials; inclusion of vulnerable groups)]